# Experimental Investigations into Earthquake Resistance of Steel Frame Retrofitted by Low-Yield-Point Steel Energy Absorbers

**Jianhua Shao [1,2,*], Kai Wang [1], Sakdirat Kaewunruen [2], Wenhua Cai [3] and Zhanguang Wang [4]**

[1]   Department of Civil Engineering, Jiangsu University of Science and Technology, Zhenjiang 212003, China
[2]   Department of Civil Engineering, School of Engineering, University of Birmingham, Birmingham B15 2TT, UK
[3]   Department of Mechanics and Structural Engineering, Yancheng Institute of Technology, Yancheng 224051, China
[4]   Department of Civil Engineering, Kaili University, Kaili 556011, China
[*]   Correspondence: shaojianhua97@163.com; Tel.: +86-15606106529

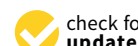

**Featured Application: A novel earthquake-energy absorber using extremely low-yield-point steel applicable to buildings' steel frame is proposed. It can significantly improve the seismic performance of buildings and infrastructures. Essentially, nonlinear dynamic behaviors and corresponding failure modes have been identified by conducting rigorous shaking table tests. The insights derived from this work will guide the advanced design of the new, innovated energy absorber.**

**Abstract:** This paper is the world's first to highlight an experimental investigation into the earthquake responses of a steel frame retrofitted by novel metallic bending energy absorbers made of low-yield-point steel with the yield strength of approximately 100 MPa. New results have been achieved by conducting comprehensive shaking table tests on a quarter-scaled model of a two-story, one-span building structure subjected to incremental intensity levels of input earthquake records. The detailed information of the specimens, material properties, monitoring sensors, and dynamic loading mechanisms has been presented. The experimental results in terms of seismic phenomena, dynamic characteristics, acceleration, inter-story drift ratios, and strain distributions are also analyzed by the data collected from a wide range of sensors. It is found that the seismic failure of the specimens depends largely on the energy absorbers, which dissipate the majority of seismic input energy in order to prevent the parent steel frame from being damaged by a severe earthquake. In addition, the retrofitted structure sufficiently satisfies the design criteria considering allowable drift limits under both frequent and rare earthquakes. This indicates the influential role of the novel low-yield-point absorber, in that the overall seismic performance of the retrofitted structure can be improved adequately for survival in high-intensity seismic fortification areas.

**Keywords:** steel frame; low-yield-point steel; energy absorber; shaking table test

## 1. Introduction

Earthquakes are always seriously taken into account when designing and constructing an engineering structure such as a high-rise building, a long-span railway bridge, a sport arena, etc. [1–6]. The influence of an earthquake on building structures can tremendously cause the losses of either lives or investments. Any landmark engineering structure can be destroyed in an instant because of the occurrence of extreme motions caused by natural disasters such as earthquakes and strong winds, or even by high-speed traffic motions [7,8].

In a traditional seismic design of building structures, an earthquake energy input is designed to be dissipated by the plastic deformation of some significant structural members such as frame columns, beams, joints, and braces. The energy dissipation mechanism may cause some structural damages or a total collapse subjected to earthquake motions [9]. In order to alleviate the risk of seismic damages to urban infrastructures, many scholars have engaged in the research frontier of earthquake engineering and have continuously developed various technologies to improve the structural safety of a building under severe earthquakes. A passive energy dissipation system has been widely used in seismic regions to absorb the input energy of earthquake excitations. In addition, a structural seismic control is commonly adopted to change or adjust the dynamic characteristics of the structure by installing special devices (e.g., seismic isolation bearing [10–12]), some other structural modifications (e.g., energy-dissipation braces and joints [13–17], fluid viscous energy absorber [18–20], and metallic energy absorber [21–24]), or additional substructures (e.g., tuned mass energy absorber [25–27]) in a certain part of the structure. These devices, which require no additional energy to operate, generate a controlled counterbalance force and provide the ability to improve energy dissipation in structural systems [28]. On the other hand, a passive energy dissipation device can be used as a special structural element that, when incorporated into a structure, absorbs or consumes a significant portion of the input energy. Thereby it reduces energy dissipation demand on primary structural members and minimizes possible structural damage.

Engineers are constantly looking for new materials that increase the seismic performance of a building, and one of the most effective materials that have been found in recent decades is the low-yield-point (LYP) steel. The LYP steel possesses extremely low yield strength, low yield ratio, and high elongation capacity much larger than that of conventional structural steel. It also exhibits excellent strain hardening characteristics when subjected to cyclic loads [29–31]. With a low yield ratio, the structure that utilizes LYP steel can redistribute the inelastic stress easily and provides a larger plastic zone. Due to its superior deformation capacity, researches on the application of LYP steel in building structures in the forms of shear walls can be found [32–34]. The LYP steel is also frequently adopted in hysteretic metallic energy absorbers. The advantages of LYP steel energy absorbers as a novel kind of passive energy dissipation device include simple conformation, stable hysteretic performance, low cost, and explicit early-warning mechanism under earthquake action. Significant attention has been paid to the development as well as the seismic evaluation of LYP steel energy absorbers, so as to remarkably improve the performance by decreasing any damage of the parent structure [35–41]. The LYP steel hysteretic energy absorbers applied in the buildings are designed to reduce the dynamic responses of drift and stress, which utilize the considerable plastic deformation to absorb the input energy under earthquakes or other natural forces.

Previous researchers carried out significant work on the hysteretic seismic performance of LYP steel energy absorbers, which promote the application and development for building systems. However, their researches are restricted to focus on the static or cyclic loading tests and finite-element numerical simulations at a component level of LYP steel energy absorbers. Any new study on time-history dynamic seismic performance of an overall structure combined with additional energy absorbers is rarely carried out, let alone the shaking table tests of structures.

In this paper, a novel type of metallic bending panel energy absorber fabricated by using low-yield-point steel with a yield strength of approximately 100 MPa is presented. The quarter-scaled steel frame specimen retrofitted by LYP steel energy absorbers is designed, and the peak ground accelerations of three seismic records, including El Centro, Kobe and Taft waves with different motion characteristics, are scaled to a series of amplitude modulations. In order to investigate the nonlinear seismic behaviors of the structure, such as the failure modes, dynamic properties, acceleration responses, inter-story drift ratios, and strain distribution, shaking table tests are performed on the specimen, subjected to the increasing intensity levels of earthquake records. The comprehensive test results provide novel and practical insights and act as a new technical reference on the engineering application and theoretical analysis for this kind of metallic energy absorber.

## 2. Specimen Design and Test Method

### 2.1. Specimen Design

Due to the limitation of bearing capacity provided by our shaking table facility, a quarter-scale spatial structural model of a single-span, two-story steel frame retrofitted by low-yield-point steel energy absorbers was designed in accordance with the Chinese Code for Seismic Design of Buildings (GB 50011-2010) [42]. The materials used for the test model were identical to those of the prototype structure, thereby indicating that the similitude ratio of the elastic modulus is $S_E = 1$. The scaling factor of the acceleration was thus taken as $S_\alpha = 2.5$ according to the maximum acceleration of ground records provided by the shaking table in laboratory conditions. The remaining similitude ratios of the mechanical parameters are listed in Table 1. The experimental results of the scaled model were then converted by the corresponding similitude ratios to obtain the dynamic responses of the prototype structure.

**Table 1.** Similitudes of the test model.

| Physical Quantities | Dimensions | Similarity Ratios | Physical Quantities | Dimensions | Similarity Ratios |
|---|---|---|---|---|---|
| Length | L | $S_L = 1/4$ | Time | T | $S_T = 0.315$ |
| Young's modulus | $FL^{-2}$ | $S_E = 1$ | Poisson ratio | 1 | $S_\nu = 1$ |
| Stress | $FL^{-2}$ | $S_\sigma = 1$ | Stiffness | $FL^{-1}$ | $S_k = 1/4$ |
| Mass | $FT^2L^{-1}$ | $S_m = 1/40.3$ | Frequency | $T^{-1}$ | $S_f = 3.175$ |
| Acceleration | $LT^{-2}$ | $S_\alpha = 2.5$ | Strain | 1 | $S_\varepsilon = 1$ |

The plane size of a specimen with a height of 0.8 m at each story, shown in Figure 1, was 1.2 × 1.2 m considering the limited size and loading capacity of the shaking table. Hot-rolled H-shaped steel with type specification of H 100 × 100 × 6 × 8 mm was adopted for all the frame columns (FC), and the I-shaped steel of I10 was used for all the main beams (MB) and secondary beams (SB). The LYP steel energy absorbers were connected to the lower flange of the main frame beam by equal-leg, double-angle steel bracing (EDB) with a type specification of 2L 45 × 3 mm (Figure 1c). The low yield strength steel was then used for the four middle plates with a thickness of 8 mm welded to the two upper and lower ordinary steel plates with a 20 mm thickness in the metallic energy absorbers, whereas ordinary Q235B steel was applied in all other steel components. The additional mass blocks with a weight of 2 tons were mounted on each story of the specimen to simulate the weight distribution of the prototype structure accurately. Taking the mass blocks into consideration, the total weight of this specimen including the mass of the steel columns, beams, bracings, metallic energy absorbers, and connecting plates, was approximately 4.41 tons.

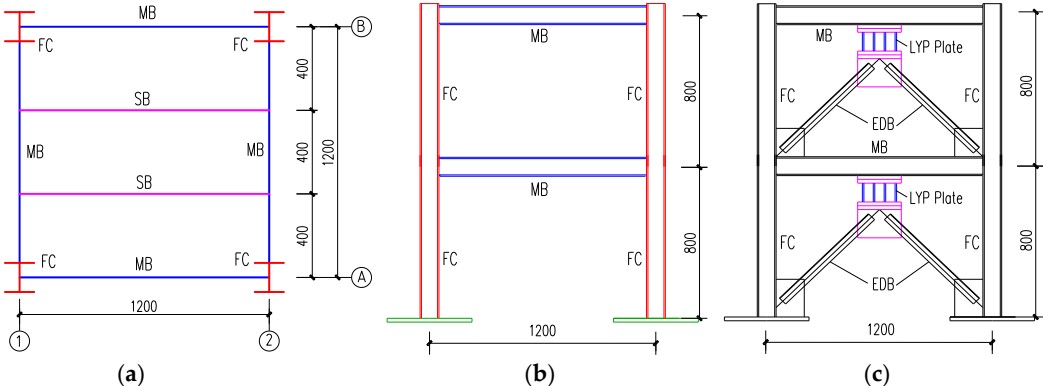

**Figure 1.** Specimen size (unit: mm). (**a**) Plane layout; (**b**) elevation at first or second axis; (**c**) elevation at A or B axis.

The bottom of the steel column in Figure 2 was strengthened by multiple stiffeners to achieve sufficient strength and stiffness, in order to prevent the column base from any premature formation of local buckling or plastic hinge during the tests. A cantilever splicing section located at the weak axis of the frame column was connected with the main frame beam by the full penetration butt weld to ensure the rigid beam-column joint, as shown in Figure 3. The energy absorber in Figure 4 was designed to bend and yield from the middle at first and then spread to the two ends during the test, which was composed of four LYP steel plates with the middle section smaller than the upper and lower end sections. The arc transition at the cross-sectional change was then adopted to avoid the premature failure caused by the local stress concentration.

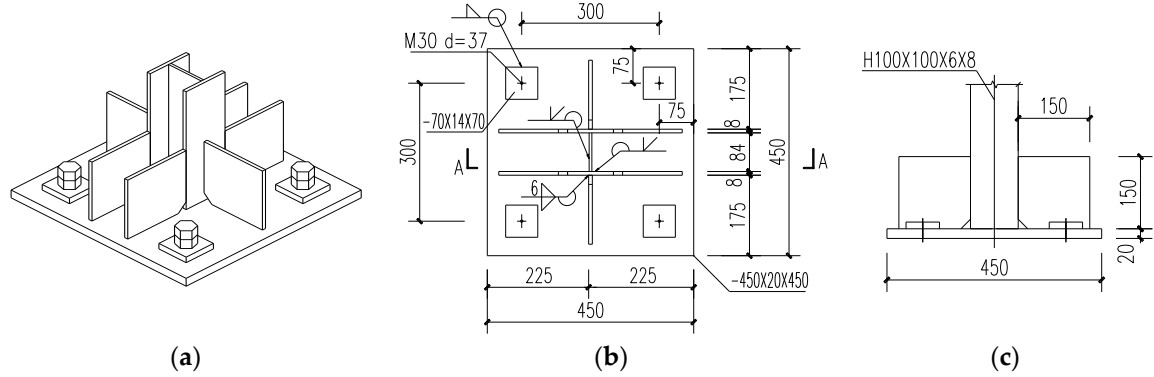

**Figure 2.** Detailed connections at the bottom of columns (unit: mm). (**a**) 3D; (**b**) plane; (**c**) section A–A.

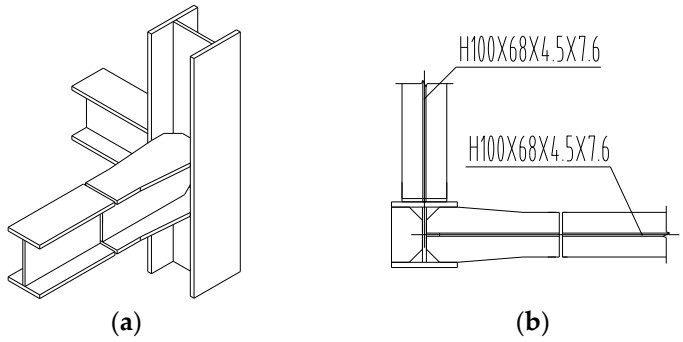

**Figure 3.** Beam–column joint. (**a**) 3D; (**b**) plane.

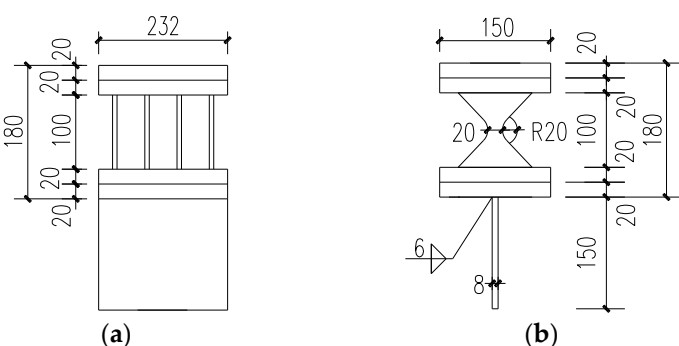

**Figure 4.** Energy absorbers (unit: mm). (**a**) Front; (**b**) side.

*2.2. Material Property Test*

The chemical compositions of test materials are given in Table 2. The content of carbon and manganese in low-yield-point steel was much lower than that in ordinary steel. The lower the carbon content, the better the plasticity, cold bending performance, weldability, and corrosion resistance of steel, but the lower the strength of steel. Manganese can significantly improve the strength of steel,

but reduce the weldability of steel. These indicate that the yield strength of low-yield-point steel is lower, whereas the plasticity and weldability are better than that of ordinary steel.

**Table 2.** Chemical compositions of test materials (unit: %).

| Test Materials | C | Si | Mn | P | S | N |
|---|---|---|---|---|---|---|
| Low-yield-point steel | 0.01 | 0.06 | 0.08 | 0.007 | 0.01 | 0.0024 |
| Ordinary steel | 0.16 | 0.15 | 0.33 | 0.028 | 0.021 | 0.045 |

The uniaxial tensile test on steel coupons had been carried out to obtain fundamental mechanical properties of the materials including the yield strength, ultimate tensile strength, modulus of elasticity, elongation, and yield ratios. The typical steel coupons designed as the shape of the plate shown in Figure 5 for the material tensile test were cut from the base metals. Table 3 presents the detailed sizes of different geometric parameters about the coupons, and three tensile coupons with the actually measured sizes listed in Table 4 were tested for the LYP steel with the nominal yield strength of 100 MPa.

**Table 3.** Sizes of the coupon (unit: mm).

| Parameters | Thickness $a$ | Gauge Width $b$ | Gauge Length $L_0$ | Parallel Length $L_c$ | Transition Arc Length $r$ | Clamp Width $d$ | Overall Length $L$ |
|---|---|---|---|---|---|---|---|
| Sizes | 8 | 25 | 80 | 110 | 25 | 45 | 310 |

**Table 4.** Measured sizes of LYP steel coupons (unit: mm).

| Number | M1 | M2 | M3 |
|---|---|---|---|
| Measured thickness $a$ | 7.56 | 7.57 | 7.57 |
| Measured width $b$ | 25.02 | 25.01 | 24.98 |

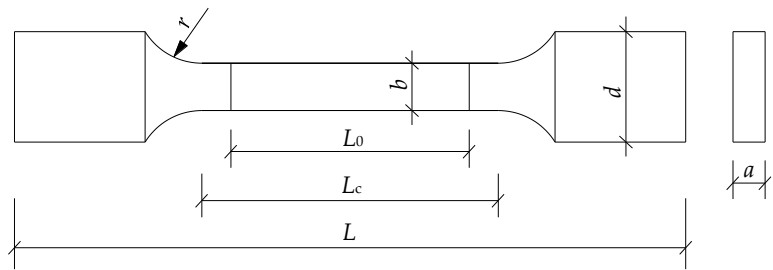

**Figure 5.** Fabricated coupon.

These steel coupons for tensile tests were respectively installed on the electro-hydraulic universal material-testing machine, which can automatically record the maximum load, yield strength, and ultimate strength at the end of the test (Figure 6). The tensile coupons of LYP steel were observed to stretch considerably long, from the initial elasticity to the yield and then to the strain hardening stage until the necking fracture. The cross-sectional area was reduced to a sufficiently small section, and the fractured surface was uneven and jagged after tensile fracture of LYP steel coupons, as shown in Figure 7, which indicates that this LYP steel exhibits superior ductile performance.

Table 5 presents the different mechanical behaviors of low-yield-point and ordinary steel specimens under quasi-static tensile tests. In comparison, the ratios of yield strength, ultimate tensile strength, and yield ratio of LYP steel to Q235 steel were approximately 0.41, 0.64, and 0.63, respectively, whereas the elongation of LYP steel was approximately 1.36 times that of Q235 steel. The actual engineering stress–strain curves of LYP steel could then be calculated and achieved based on the test results, as shown in Figure 8.

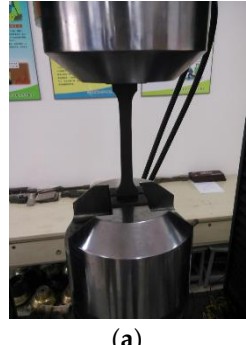 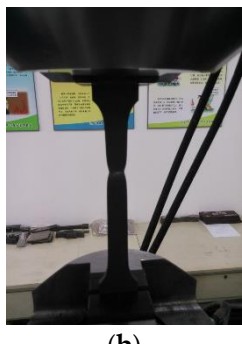

(**a**)                    (**b**)

**Figure 6.** Material tensile test. (**a**) Installation; (**b**) fracture.

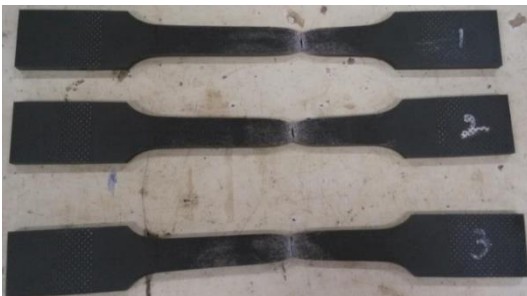

**Figure 7.** Fractured LYP steel coupons.

**Table 5.** Results of the material test.

| Specimen | | Maximum Load | Yield Strength | Tensile Strength | Modulus of Elasticity | Elongation | Yield Ratio |
|---|---|---|---|---|---|---|---|
| Material | Number | $F$ (kN) | $f_y$ (MPa) | $f_u$ (MPa) | $E$ (MPa) | (%) | $f_y/f_u$ |
| Low-yield-point steel | M1 | 47.42 | 100.59 | 250.68 | 198,639 | 46.36 | 0.40 |
| | M2 | 47.13 | 94.74 | 248.96 | 208,734 | 48.18 | 0.38 |
| | M3 | 47.04 | 90.76 | 248.76 | 200,974 | 48.18 | 0.36 |
| Ordinary steel (Q235) | | 74.26 | 234.91 | 391.21 | 200,152 | 35.10 | 0.60 |

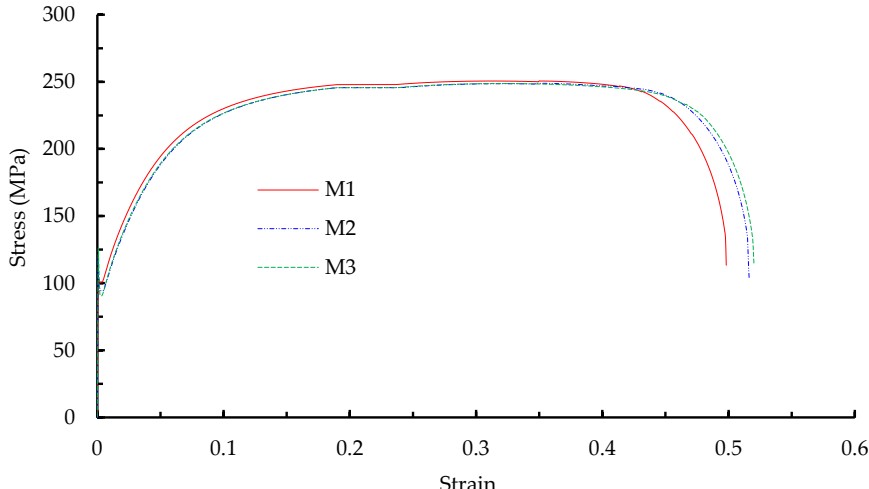

**Figure 8.** Stress–strain curves of the LYP steel.

The characteristics of LYP steel regarded as an ideal material for energy absorption and its advantages in engineering applications can be summarized as follows: (i) the yield strength of LYP steel is much lower than that of ordinary steel, which will enable metallic energy absorbers made of LYP steel to form the plastic deformation, prior to other structural members, for energy dissipation when

subjected to earthquake load or intensive wind load; (ii) the yield strength is less than 0.5 times ultimate strength, and the much prolonged strain-hardening process accompanied by the accumulation of plastic deformation will contribute to the seismic-resisting capacity and guarantee the required lateral stiffness under earthquake action; and (iii) the elongation at break is approximately 48% and the favorable plastic capacity without deterioration will ensure the superior structural ductility. Attributable to its high plastic deformation capacity, the LYP steel has thus been widely applied in seismic engineering.

### 2.3. Fabrication and Installation of the Specimen

All the steel components in the specimen were fabricated and connected in the factory. The LYP steel energy absorbers after the completed fabrication, as shown in Figure 9, was welded to the lower flange of the frame beam through the end plate, and also welded together with the equal-leg, double-angle steel bracing through the connecting plate. The welded connection was adopted in the beam–column joints to possess a rigid stiffness. A bottom plate was welded to the end of the steel column, which was firmly attached to the shaking table surface with the high-strength bolt. A hole was reserved at the center of each floor plate for the placement and fixation of additional mass blocks.

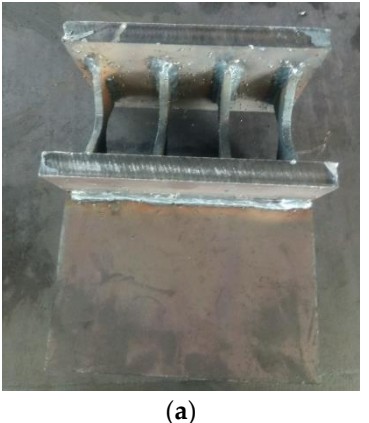
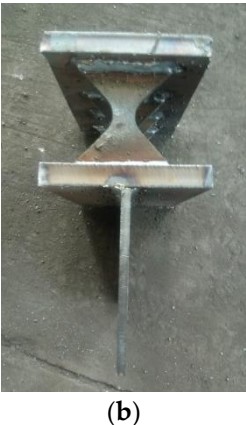

(**a**)                                      (**b**)

**Figure 9.** Manufactured LYP steel energy absorbers. (**a**) Front view; (**b**) side view.

After being transported to the structural laboratory, the whole specimen was hoisted to the shaking table surface by a bridge crane in the laboratory and fixed by the anchor bolt. Then, artificial counterweights of additional mass blocks in the form of rubber isolation bearings were deployed and bolted to the steel floor slab in order to ensure that no slipping could be occurred throughout the test, as shown in Figure 10.

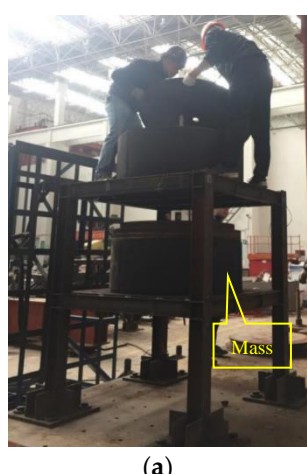
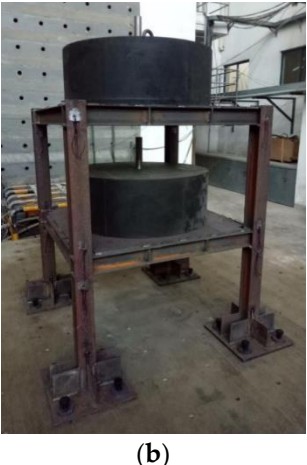

(**a**)                                      (**b**)

**Figure 10.** Installed specimen. (**a**) Placing mass block; (**b**) hoisting at the appropriate position.

### 2.4. Instrumentation

A series of shaking table tests under three seismic waves, with the scaled amplitude modulation of peak acceleration, was carried out to investigate the overall damage process of the structure subjected to the different intensity levels of earthquake records. The strains and displacements at various locations of the specimen under the action of different seismic waves were monitored to analyze the nonlinear dynamic time-history responses of the structural model during the test.

The geometric working size was 4.0 × 6.0 m for the loading test of the shaking table facility with a maximum capacity of up to 250 kN, operating frequency range 0.1–50 Hz, and maximum velocity of ±6 mm/s. The shaking table could produce a maximum acceleration of 3.0 g at the non-load condition and 1.5 g at the loading condition of 250 kN, with a maximum displacement of ±250 mm, as shown in Figure 11.

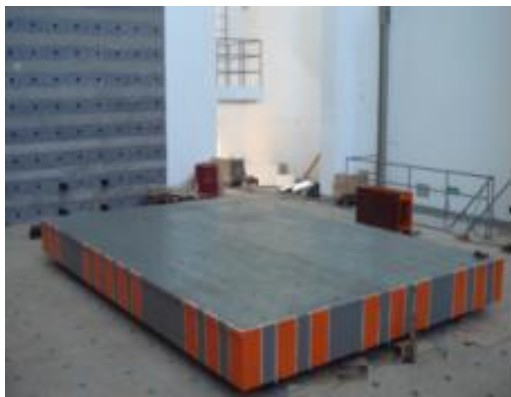

**Figure 11.** Shaking table.

The resistance strain gauges were adopted to investigate the strain variance of steel members during this test (as shown in Figure 12a). The displacement sensor (type: YHD-50) with a range of ±50 mm and another jacking displacement meter (type: TST-250) with a range of ±250 mm were mounted on the model to monitor the inter-story drift ratios (see Figure 12b,c). The corresponding dynamic signal processing system was used for data acquisition and analysis (see Figure 12d).

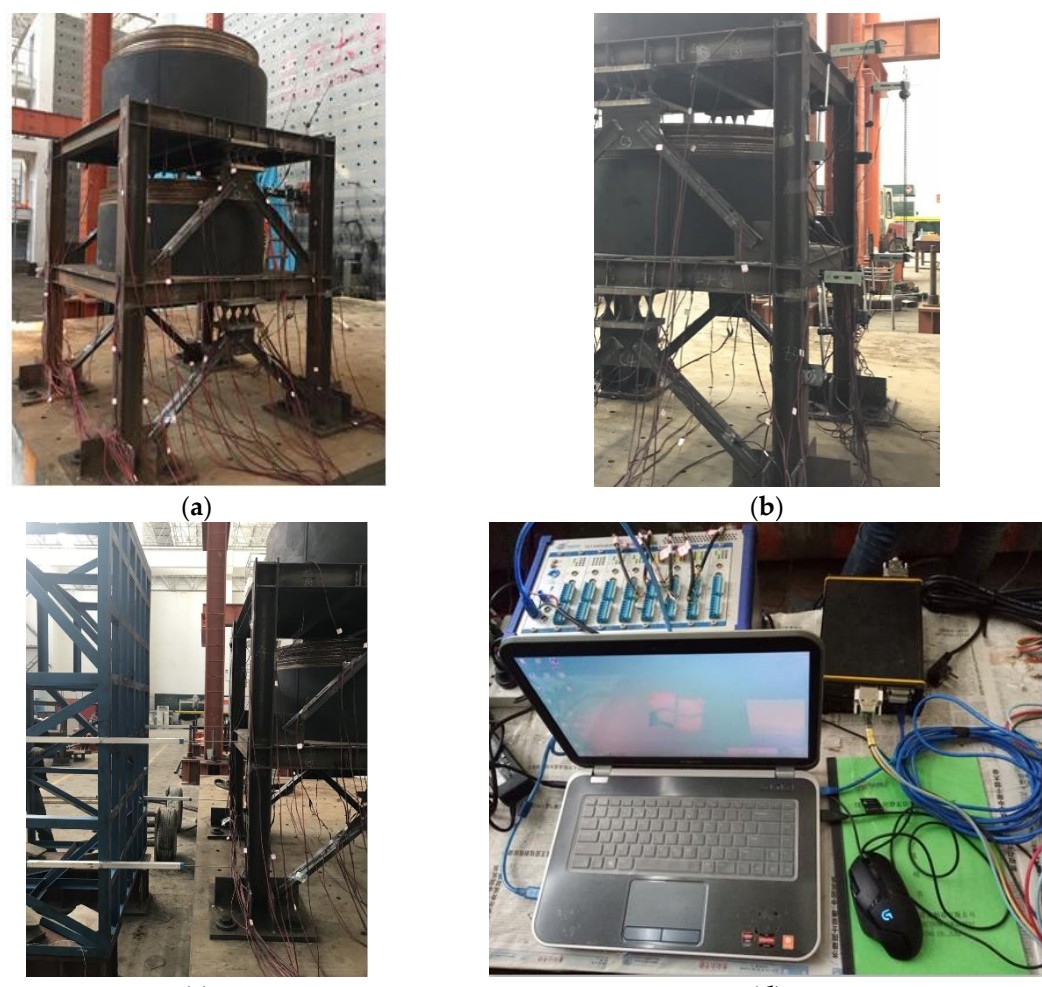

**Figure 12.** Measuring devices of the specimen. (**a**) Connection of strain gauges; (**b**) YHD sensors; (**c**) TST sensors; (**d**) data acquisition instrument.

*2.5. Layout of Monitoring Points*

The appropriate positions of measurement were defined to capture the overall time-history dynamic responses of the test model during seismic loadings, as well as any local effects such as yielding of LYP steel energy absorbers. Figure 13 shows the arrangement of monitoring points where the strain gauges and displacement sensors for the measurement of each component are placed. The strain gauges used for acquiring the maximum stress to monitor the damaged condition of frame members were respectively glued on the flanges at both ends of columns or beams. They were located at the area where the stress induced by bending moment under the horizontal ground motion was much higher than that at the middle or other parts. The measurement points of strains in the energy absorbers were arranged at the center of each LYP steel plate, where the stress caused by the bending deformation under the action of horizontal shear force transferred from the frame beam was the maximum. The waterproof treatment and protection measures were performed to prevent dampness and damage after the strain gauges had been glued on the polished steel surface with the dirt removed. The YHD type of displacement sensors were fixed on the outer flange of the frame column, at the beam–column connections of each story, by a magnetic suction seat to measure the inter-story drift of the specimen. Another TST type of displacement meter was installed on the bracket by a steel wire to obtain the absolute displacement at the monitoring point.

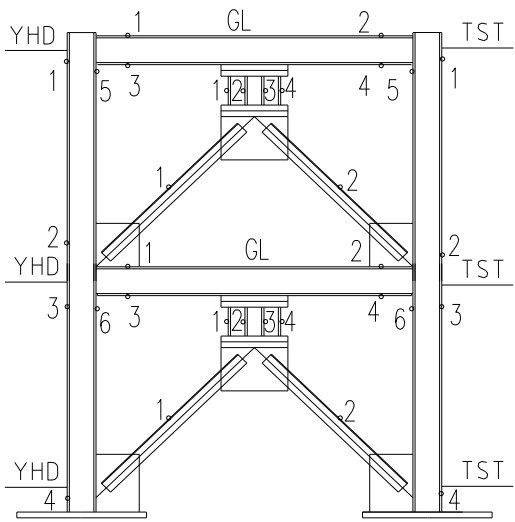

**Figure 13.** Layout of monitoring points.

## 2.6. Loading Protocol

The shaking table tests were rigorously carried out to evaluate the seismic performance of the model by the increasing earthquake excitation magnitudes. Three ground input excitations, representative of actual earthquake records, which conform to the dynamic characteristics of the standardized response spectrum put forward by GB 50011-2010, were selected: El Centro wave (Imperial Valley earthquake, 1940), Kobe wave (Hyogoken-Nanbu earthquake, 1995), and Taft wave (California earthquake, 1952). The unidirectional alternative-loading method of the seismic wave was applied to investigate the dynamic characteristics of the model under different levels of earthquake intensity in this test.

Firstly, the model was scanned by 3D white noise signals, for which the peak ground acceleration (PGA) level was limited to 0.05 g, to obtain the original dynamic characteristics, including the natural frequency and damping ratio, before applying the ground motion. Subsequently, a white noise excitation was applied to identify any variation in the dynamic characteristics of the test model after each level of adjusted PGA for the earthquake records.

Three different levels of earthquake hazard with 63.2%, 10%, and 2% probabilities of exceedance in 50 years of the design reference period are referred to as "frequent earthquake", "occasional earthquake" and "rare earthquake", respectively. Table 6 lists the loading plan consisting of 47 seismic cases during the shaking table test program. The loading-scaled PGAs of input El Centro and Taft waves were increased from 0 to 1.0 g, with each case increased by 0.1 g, whereas the Kobe wave was scaled up to 1.6 g due to the different maximum oil pressures caused by the individual dynamic characteristics of seismic waves. The adjusted PGAs were involved in all levels of earthquake fortification intensity, from frequent 7 to rare 9, for the test model presented in Table 7. The representative seismic waves and corresponding frequency spectra, derived from converting the input time domain into the frequency domain by fast Fourier transform under PGA = 0.3 g, as an example for three ground records, are shown in Figures 14 and 15, from which the dominant frequencies of 1.82, 2.54, and 1.72 Hz were obtained for the El Centro, Kobe, and Taft earthquake waves, respectively.

**Table 6.** Seismic loading plan.

| Case | Earthquake | PGA (g) | Case | Earthquake | PGA (g) | Case | Earthquake | PGA (g) | Case | Earthquake | PGA (g) |
|---|---|---|---|---|---|---|---|---|---|---|---|
| 1a | White noise | 0.05 | 4a | White noise | 0.05 | 7a | White noise | 0.05 | 10a | White noise | 0.05 |
| 1 | El Centro | 0.1 | 10 | El Centro | 0.4 | 19 | El Centro | 0.7 | 28 | El Centro | 1.0 |
| 2 | Kobe | 0.1 | 11 | Kobe | 0.4 | 20 | Kobe | 0.7 | 29 | Kobe | 1.0 |
| 3 | Taft | 0.1 | 12 | Taft | 0.4 | 21 | Taft | 0.7 | 30 | Taft | 1.0 |
| 2a | White noise | 0.05 | 5a | White noise | 0.05 | 8a | White noise | 0.05 | 11a | White noise | 0.05 |
| 4 | El Centro | 0.2 | 13 | El Centro | 0.5 | 22 | El Centro | 0.8 | 31 | Kobe | 1.1 |
| 5 | Kobe | 0.2 | 14 | Kobe | 0.5 | 23 | Kobe | 0.8 | 32 | Kobe | 1.2 |
| 6 | Taft | 0.2 | 15 | Taft | 0.5 | 24 | Taft | 0.8 | 33 | Kobe | 1.3 |
| 3a | White noise | 0.05 | 6a | White noise | 0.05 | 9a | White noise | 0.05 | 34 | Kobe | 1.4 |
| 7 | El Centro | 0.3 | 16 | El Centro | 0.6 | 25 | El Centro | 0.9 | 35 | Kobe | 1.5 |
| 8 | Kobe | 0.3 | 17 | Kobe | 0.6 | 26 | Kobe | 0.9 | 36 | Kobe | 1.6 |
| 9 | Taft | 0.3 | 18 | Taft | 0.6 | 27 | Taft | 0.9 | | | |

**Table 7.** PGA of earthquake intensity for the test.

| PGA (g) | 0.0875 | 0.175 | 0.25 | 0.35 | 0.5 | 0.55 | 1.0 | 1.55 |
|---|---|---|---|---|---|---|---|---|
| **Earthquake Intensity** | Frequent 7 | Frequent 8 | Occasional 7 | Frequent 9 | Occasional 8 | Rare 7 | Rare 8/ Occasional 9 | Rare 9 |

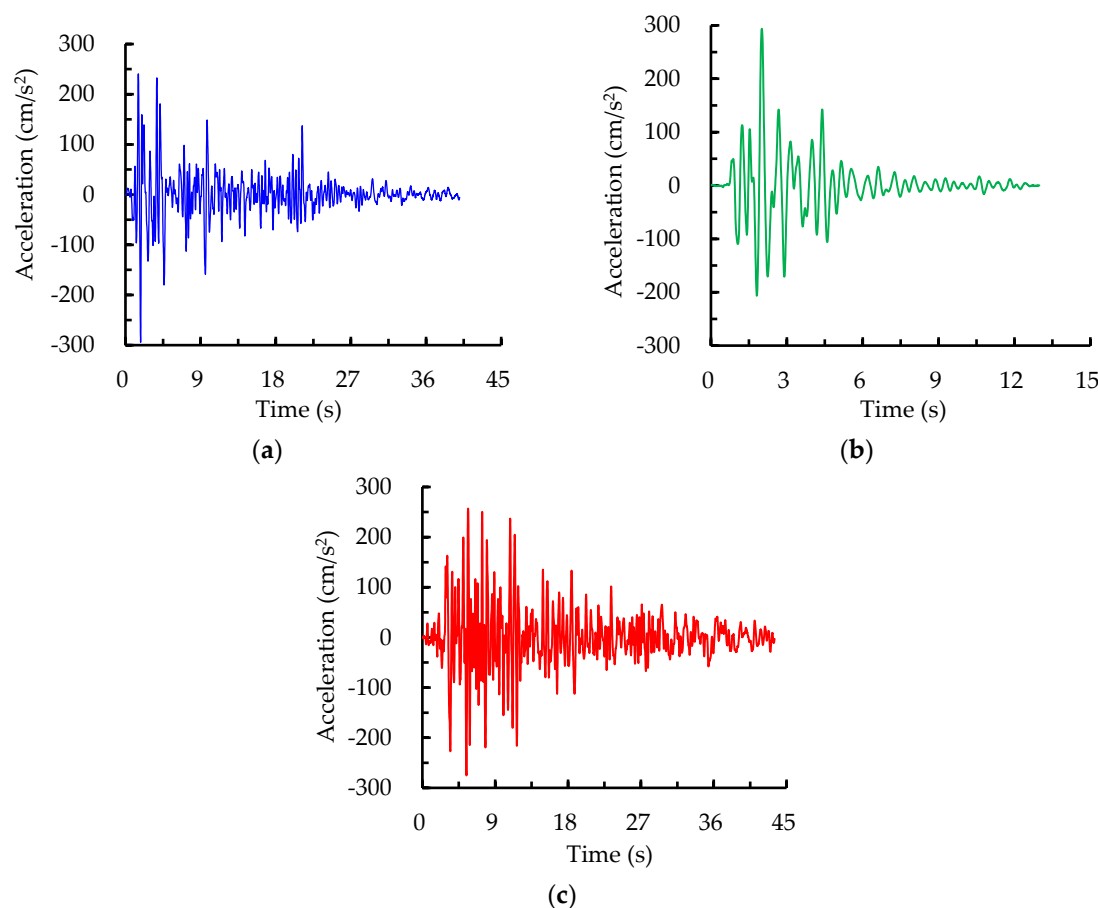

**Figure 14.** Loading earthquake waves. (**a**) El Centro wave; (**b**) Kobe wave; (**c**) Taft wave.

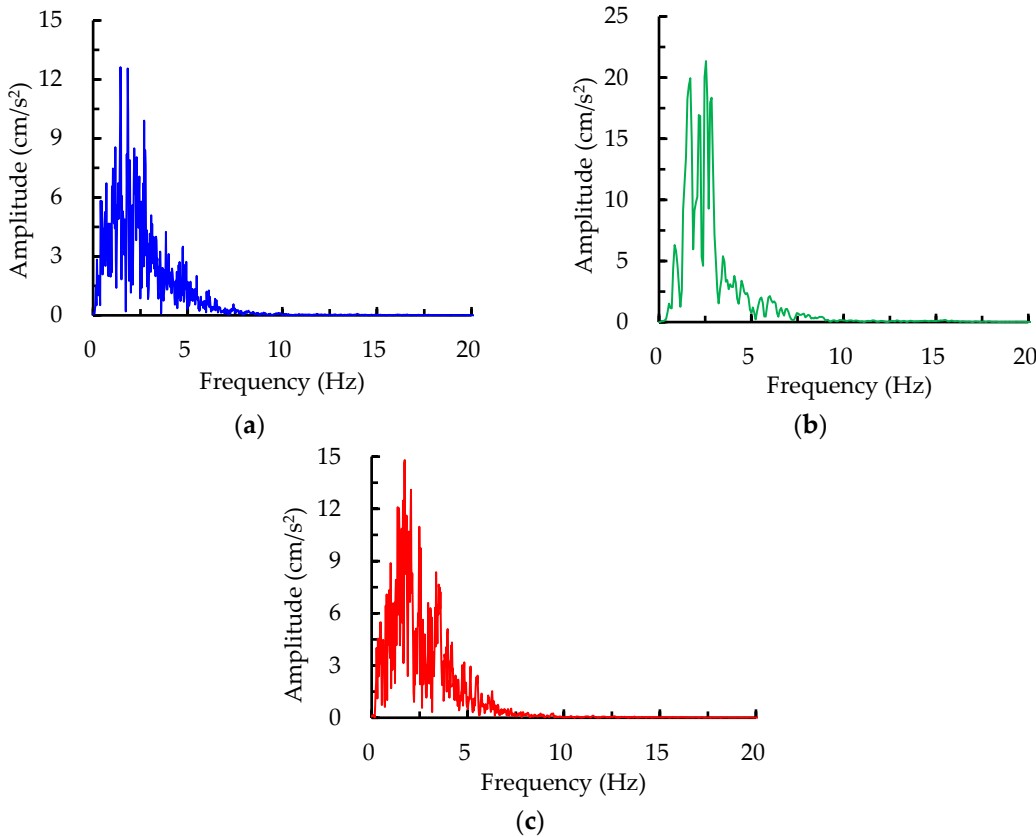

**Figure 15.** Corresponding frequency spectra of waves. (**a**) El Centro wave; (**b**) Kobe wave; (**c**) Taft wave.

## 3. Experimental Results and Discussion

### 3.1. Experimental Observations

It could be observed from the structural responses that the specimens vibrated most intensely under the input excitations of the Kobe wave, which possessed the larger vibration amplitude and stronger excitation energy, seen from Figure 15, compared to the El Centro and Taft waves when subjected to the same PGA during the shaking table tests. Before the loading case #10 with a maximum PGA of 0.4 g, no visual and acoustical evidence of all the steel components was demonstrated in the test model, and the structural performance remained in an elastic working condition during the frequent earthquake. When the PGA reached up to 0.6 g, a noticeable tremor and shaking sound could be observed from the upper additional mass block, and only a slight shaking response was found at the lower mass block. However, local or lateral buckling in the LYP steel energy absorbers or columns was not observable, thereby revealing that the test model remained elastic until such loading stage. When the PGA level of the input ground motions increased from 0.7 to 1.0 g, a significantly strong shaking response, with an extremely loud and continuous sound, occurred in the mass block. It was accompanied by a considerable amount of dust drifting down, and the overall model intensely vibrated with the increasingly visible displacement. Meanwhile, loosening of the high-strength bolts at the column bases incurred. Moreover, a small number of the displacement sensors mounted on the model broke and dropped down, and some connecting wires of strain gauges fell off (Figure 16a). When the PGA reached up to 1.3 g, the columns, beams, and lateral braces continued to work well. However, the dramatically visible and irreversible bending deformation close to the middle of the LYP steel plates, caused by the shear action, was observed (Figure 16b). This indicates that the ability to dissipate the input earthquake energy mainly relies on the out-of-plane bending mechanism of energy absorbers in any rare earthquake. When the PGA further increased from 1.4 to 1.6 g, the energy absorbers buckled severely, and the corresponding residual plastic deformation was more prominent.

In addition, the welds located at the energy absorbers and beam–column joints, where the uneven tension texture appeared on the steel surface, were slightly fractured due to the recursive tensile and compressive forces, as shown in Figure 16c,d.

Throughout the whole loading process of the tests, except for the LYP steel energy absorbers, no apparent failure occurred for the frame columns, frame beams, and lateral braces. The test model did not seriously damage and collapse until the end of the seismic loading cases. In summary, out-of-plane local buckling accompanied by significant bending deformation of LYP steel plates, and the slight fracture of fillet welds, were considered to be the governing failure modes for the test model in this research. These results clearly suggest that the seismic resisting performance of a steel structure retrofitted by additional LYP steel energy absorbers is superior. As the first line of defense in the lateral force resisting system, the energy absorbers, as seismic applications in a structure, contained adequate elastic stiffness to withstand a small earthquake or wind load. However, they performed really well in the elastic-plastic stage and exhibited excellent energy absorbability in a rare earthquake or severe wind, thereby reducing or avoiding the damage of the primary structures, such as frame columns and beams.

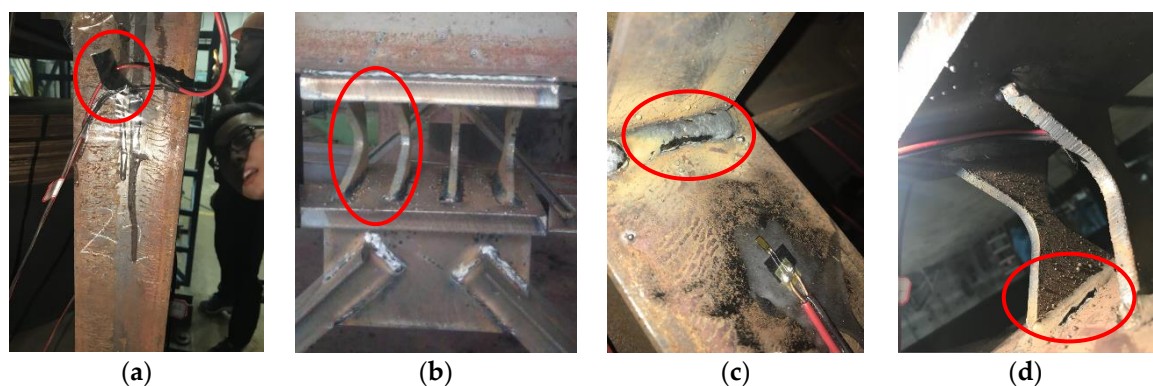

| **(a)** | **(b)** | **(c)** | **(d)** |

**Figure 16.** Test phenomena of the specimen. (**a**) Fallen wire; (**b**) deflection of energy absorbers; (**c**) fracture of the weld at the beam–column joint; (**d**) fracture of the weld at the LYP plate.

### 3.2. Natural Vibration Properties

Before and after each intensity level test, the white noise wave was utilized to sweep through the test model in order to investigate the change of its vibration properties. The structural natural frequencies of the specimen after each loading case can be obtained by analyzing the time-history acceleration responses derived from the second order differential of the measured displacement, spectrum characteristic, and the transfer function of each story under the white noise excitation. In addition, the corresponding damping ratio at a specific natural frequency can be calculated by adopting the half-power bandwidth method in Figure 17 as follows:

$$\xi = (f_2 - f_1)/(2f), \tag{1}$$

where $\xi$ is the damping ratio, $f$ is the frequency corresponding to the peak amplitude $A_{max}$ of the transfer function, $f_1$ and $f_2$ are the frequencies corresponding to the value equal to 0.707 $A_{max}$, respectively.

Table 8 presents the fundamental natural frequencies of vibration and damping ratios of the test model under the different increasing seismic excitation cases. It was found that the fundamental natural frequencies of the specimen were much higher than the dominant frequencies of three seismic waves applied as the earthquake excitations. It reveals that the occurrence of the resonance phenomenon can be avoided due to the significant differences of these frequencies. According to the expression of natural frequency $f = \sqrt{k/m}/(2\pi)$, the structural stiffness $k$ is directly proportional to the square of the frequency. The measured initial natural frequency and initial damping ratio of the specimen were 15.68 Hz and 3.17% subjected to the first loading stage of 0.1 g, respectively. Furthermore, the natural frequency remained essentially unchanged before PGA of 0.6 g, which implies that the test model

without the stiffness degradation remains completely elastic during these stages. However, the natural frequency of vibration gradually decreased with the increase of PGA from 0.7 to 1.0 g, demonstrating that the structural stiffness gradually declined, in agreement with the observed loosening of the high-strength bolts at the column bases. On the other hand, the damping ratio slightly increased during these loading cases because the damping of the structure resulted from a variety of sources, including various connection frictions and external aerodynamic frictions, except for the internal elastic material. A considerable amount of bolt connections were adopted in the column base, which was firmly attached to the vibration table surface with high-strength bolts in the test, and, in addition, all the additional mass blocks were connected by the bolts to the steel floor slab in the frame, which contributed to the increase of the damping ratio.

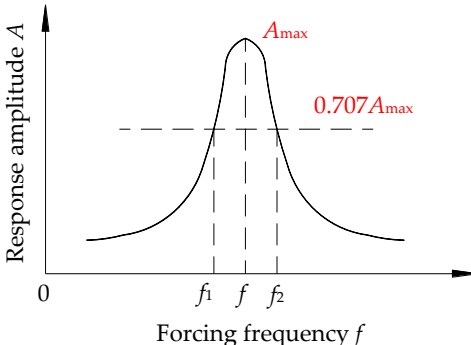

**Figure 17.** Half-power bandwidth method.

**Table 8.** Natural frequencies and damping ratios.

| Cases | 1a to 5a | 6a | 7a | 8a | 9a | 10a |
|---|---|---|---|---|---|---|
| **Natural Frequencies (Hz)** | 15.68 | 15.68 | 15.57 | 15.35 | 15.01 | 14.82 |
| **Damping Ratios (%)** | 3.17 | 3.18 | 3.19 | 3.21 | 3.23 | 3.27 |

### *3.3. Acceleration Responses*

Given the mentioned limitations of sensors, one common alternative to indirectly measure the accelerations is to carry out a second derivative of time-history displacement responses retrieved from the displacement sensors. The acceleration of amplification factor $\beta$, as an essential index, is used to reflect the structural dynamic amplification effect, because the input peak acceleration is different under each loading earthquake intensity, which can be expressed as follows:

$$\beta_i = \frac{\max(a_i(t))}{\max(a_g(t))}, \tag{2}$$

where $a_g$ and $a_i$ are the measured maximum acceleration responses at the shaking table and floor $i$, respectively.

Figure 18 shows the variation of acceleration amplification factor $\beta$ with the increasing input PGA from weak to strong. It could be seen that the measured $\beta$ of the test model under the Taft wave was the largest compared to the other earthquake motions, and all acceleration amplification factors at the second floor were significantly higher than those at the first floor subjected to the three earthquake records. The $\beta$ value was observed to sharply increase at each acceleration level within a PGA of less than 0.3 g, before frequent earthquake intensity 9, and then reached a maximum, whereas, the value steadily decreased with the increase of PGA from 0.3 to 1.0 g. When the input PGA reached 1.0 g of the rare earthquake intensity 8 for the test model, the acceleration amplification factors at the second floor were 2.3, 2.42, and 2.63, respectively, subjected to the El Centro, Kobe, and Taft waves, while those at the first floor were 1.4, 1.32, and 1.78, respectively. In conclusion, $\beta$ exhibited a sharp increase before a

certain acceleration level and then gradually decreased with the further increasing acceleration owing to the structural stiffness degradation.

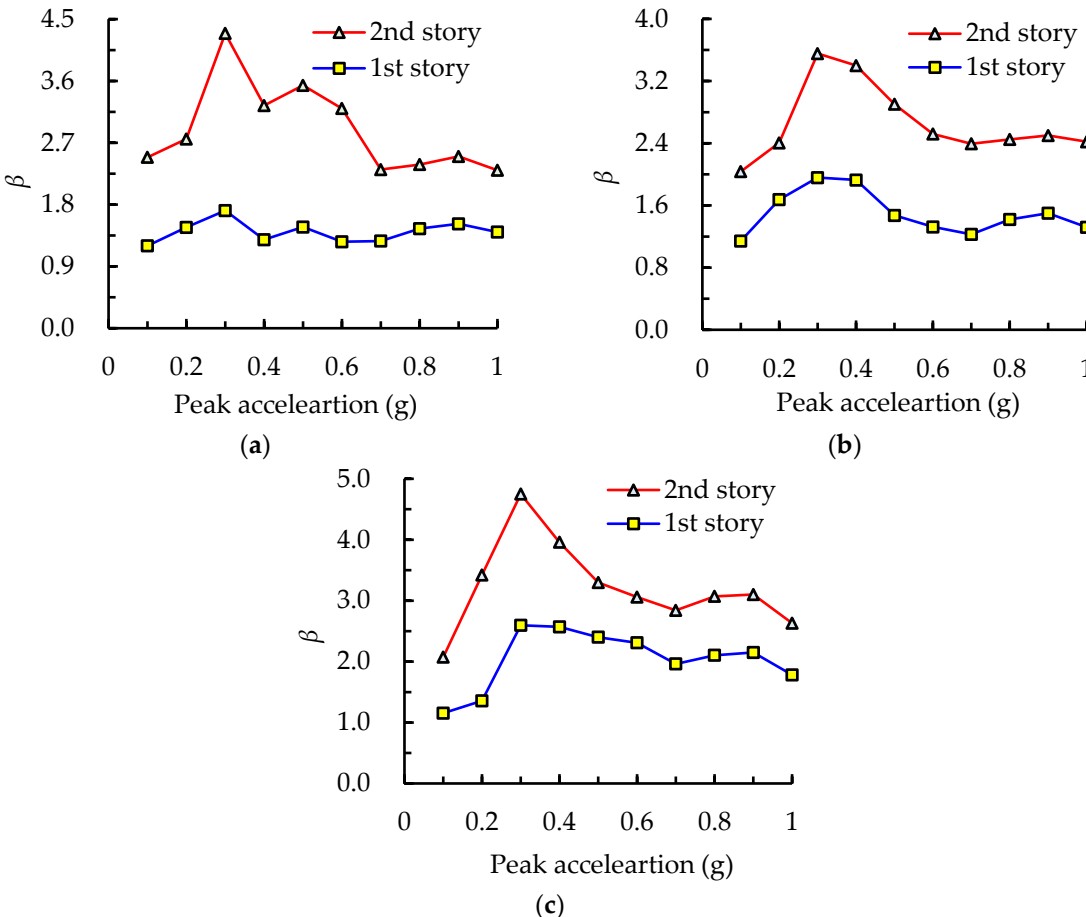

**Figure 18.** Acceleration amplification factor. (**a**) El Centro wave; (**b**) Kobe wave; (**c**) Taft wave.

*3.4. Displacement Responses*

The maximum relative displacements at each story and inter-story drift ratios of the specimen were derived from the data collected by displacement sensors installed at each floor for the different earthquake intensity levels. Only the displacement data within the PGA of 1.0 g were recorded because one part of the displacement sensors mounted on the steel column fell down when PGA reached 1.0 g. The collected time-history inter-story displacements of the test model at the bottom story under the PGA of 0.5 g, as an example for three seismic waves, are shown in Figure 19. Although the PGA was scaled to the same level for different seismic waves, the peak inter-story responses of the specimen subjected to the El Centro, Kobe, and Taft waves were still inconsistent and reached 0.53, 0.64, and 0.6 mm, respectively. This indicates that the influence of different seismic waves, possessing individual dynamic characteristics, on the structural performance varies from each other.

The measured maximum inter-story drift ratios of the specimen, which are obtained by dividing the relative lateral displacements of two adjacent floors by the corresponding story height, for three seismic waves under different earthquake intensity levels are summarized in Table 9. The drift ratios at each story exhibited a gradually increasing trend with PGA growth. The average peak drift ratio of the test model subjected to a frequent earthquake (PGA = 0.35 g) was between 1/2310 and 1/1742, which was far less than the elastic limit value of 1/250 for the multi-story steel structure given by the GB 50011-2010. This indicates that the test structure meets the specific seismic fortification goal of "no damage under minor earthquakes". Furthermore, the corresponding value subjected to a rare

earthquake (PGA = 1.0 g) was 1/233, indicating a satisfactory performance within the elastic-plastic allowable limit value of 1/50 described in this code. This also indicates that the structure meets another required goal of "no collapsing under strong earthquakes".

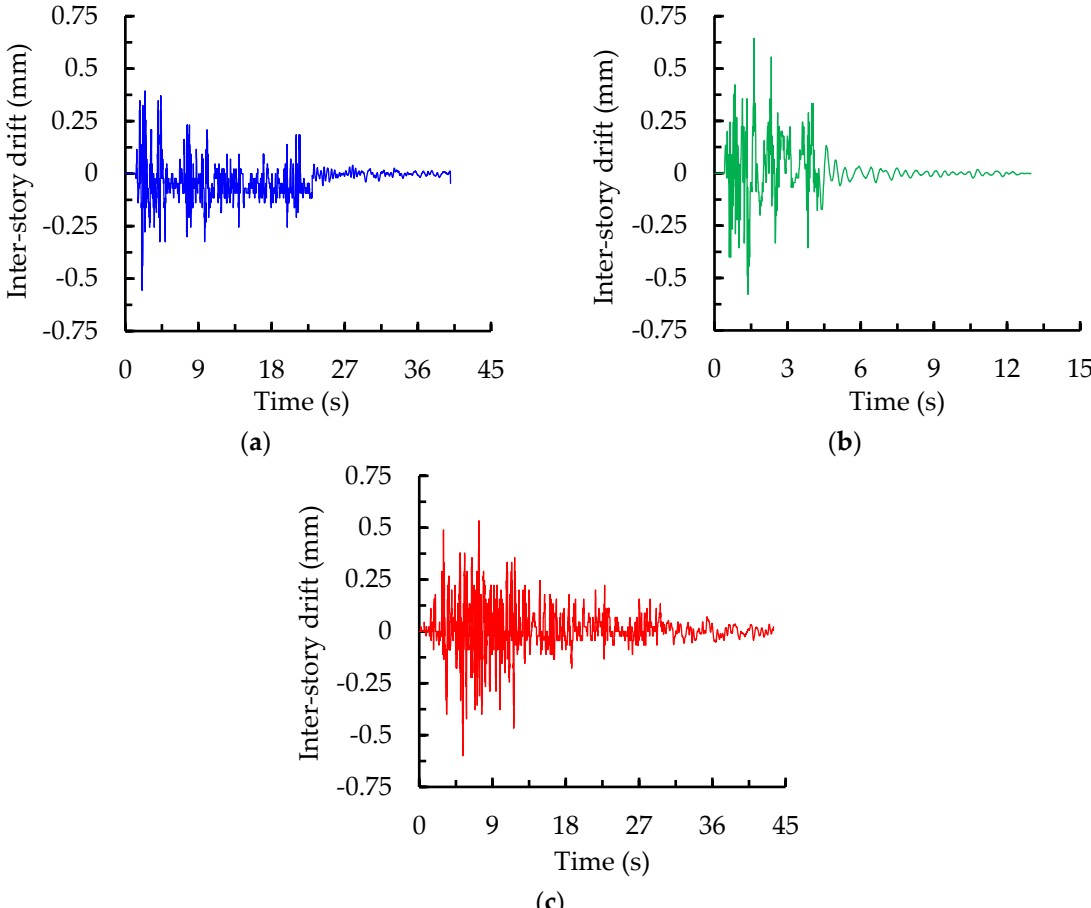

**Figure 19.** Inter-story drifts at the bottom floor under PGA of 0.5 g for three seismic waves. (**a**) El Centro wave; (**b**) Kobe wave; (**c**) Taft wave.

The above results clearly reveal that the test model sufficiently satisfies the design requirements of the seismic code and possesses a significantly high structural safety reserve even when subjected to a strong earthquake, which is attributable to the abundant lateral stiffness and remarkable energy dissipation of the structure provided by the energy absorbers installed on the inner braces. It can be concluded that the prototype structure, as a novel horizontal lateral force-resisting system, is highly applicable to high-intensity seismic fortification areas.

Figure 20 presents the variation trends of inter-story drift ratios at the top and bottom floors under different intensity levels of three seismic waves. It was found that the differences of drift ratios between the two stories under the lower intensity, before PGA < 0.7 g, were quite small; however, by further increasing the intensity level of the input seismic wave, these differences became more significant, and the measured drift ratios at the bottom story were dramatically higher than those at the top story. This indicates that the structural stiffness and strength at the bottom story are more flexible compared to the top story under the high-intensity level. In conclusion, it is recommended that the steel members at the lower stories be appropriately strengthened, by increasing the cross-sectional areas or adding more LYP plates in energy absorbers, to assure the uniform distribution of lateral stiffness along the vertical height direction for all stories, in order to avoid the occurrence of weak floors when performing the seismic design.

**Table 9.** Maximum inter-story drift ratios under various intensity levels.

| PGA (g) | Earthquake | Bottom θ | Bottom Average | Top θ | Top Average | PGA (g) | Earthquake | Bottom θ | Bottom Average | Top θ | Top Average |
|---|---|---|---|---|---|---|---|---|---|---|---|
|  | El Centro | 1/7182 |  | 1/10505 |  |  | El Centro | 1/1125 |  | 1/1381 |  |
| 0.1 | Kobe | 1/6167 | 1/6322 | 1/8003 | 1/8445 | 0.6 | Kobe | 1/1058 | 1/981 | 1/1115 | 1/1160 |
|  | Taft | 1/5775 |  | 1/7402 |  |  | Taft | 1/818 |  | 1/1036 |  |
|  | El Centro | 1/3560 |  | 1/5103 |  |  | El Centro | 1/972 |  | 1/1208 |  |
| 0.2 | Kobe | 1/3185 | 1/3181 | 1/4518 | 1/4568 | 0.7 | Kobe | 1/734 | 1/787 | 1/853 | 1/1004 |
|  | Taft | 1/2872 |  | 1/4177 |  |  | Taft | 1/705 |  | 1/1012 |  |
|  | El Centro | 1/2769 |  | 1/3207 |  |  | El Centro | 1/480 |  | 1/644 |  |
| 0.3 | Kobe | 1/2271 | 1/2310 | 1/2816 | 1/2915 | 0.8 | Kobe | 1/437 | 1/433 | 1/580 | 1/595 |
|  | Taft | 1/2011 |  | 1/2760 |  |  | Taft | 1/391 |  | 1/567 |  |
|  | El Centro | 1/2015 |  | 1/2900 |  |  | El Centro | 1/339 |  | 1/518 |  |
| 0.4 | Kobe | 1/1812 | 1/1742 | 1/2231 | 1/2523 | 0.9 | Kobe | 1/295 | 1/303 | 1/491 | 1/494 |
|  | Taft | 1/1484 |  | 1/2525 |  |  | Taft | 1/281 |  | 1/475 |  |
|  | El Centro | 1/1437 |  | 1/1611 |  |  | El Centro | 1/250 |  | 1/432 |  |
| 0.5 | Kobe | 1/1241 | 1/1332 | 1/1600 | 1/1550 | 1.0 | Kobe | 1/245 | 1/233 | 1/367 | 1/412 |
|  | Taft | 1/1333 |  | 1/1450 |  |  | Taft | 1/209 |  | 1/446 |  |

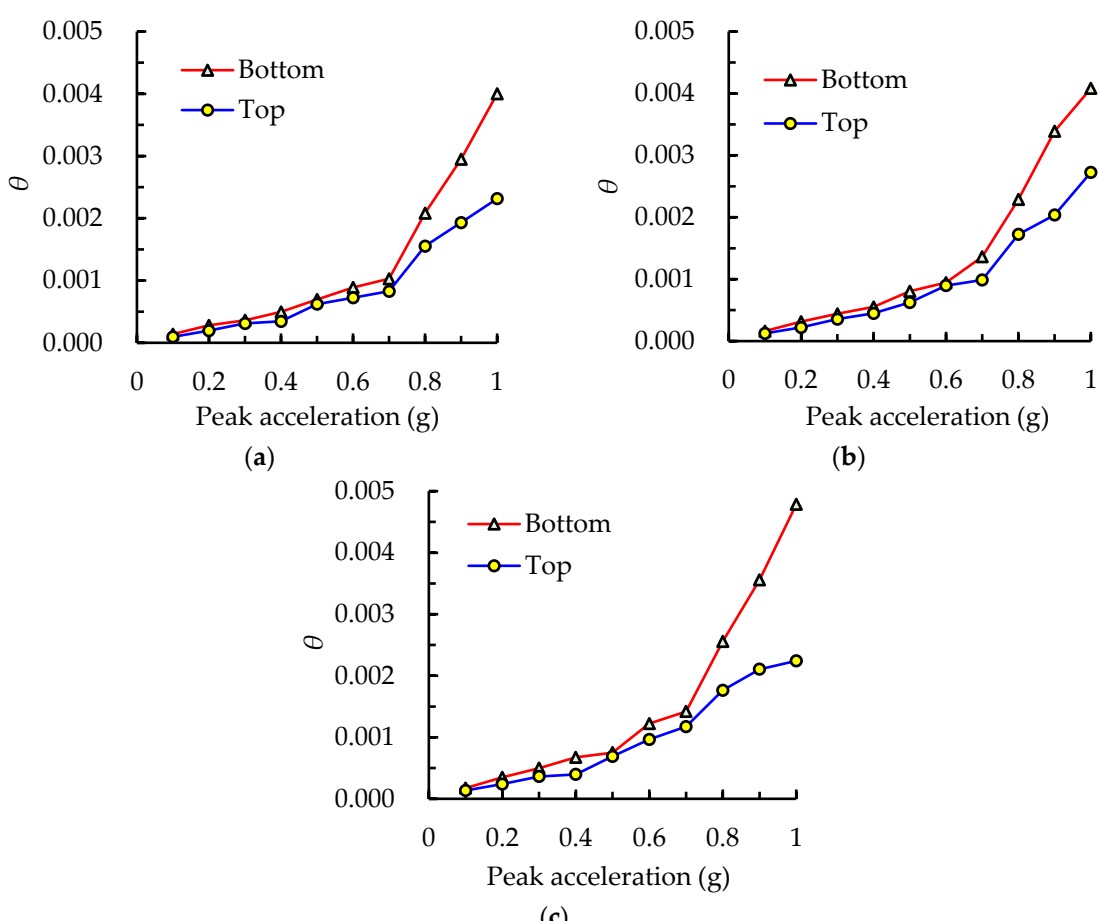

**Figure 20.** Maximum inter-story drift ratios versus loading peak accelerations for three seismic waves. (**a**) El Centro wave; (**b**) Kobe wave; (**c**) Taft wave.

### 3.5. Strain Responses of Frame

Figure 21 presents the time-history curves of strains at the bottom of the column as an example under the rare earthquake intensity 8 with PGA of 1.0 g for three seismic waves. The strain envelope values subjected to the El Centro, Kobe, and Taft waves were 318, 340, and 298 με, respectively, which were less than the yield strain of 1140 με deduced from the measured yield strength of Q235 steel in Table 5, indicating that the frame column was still in the elastic stage under a rare earthquake. The measured maximum strain derived from the Kobe record with stronger excitation energy seen

from Figure 15 was more significant compared to that from the other two records. The strain result clearly demonstrates that the strain responses are inconsistent, even when PGA is scaled to adjust at the same level for different ground excitations, due to the individually essential factors of motion including the duration and spectrum characteristics.

The measured strain responses of the model at each monitoring point under three earthquake records were obtained by the strain gauges, glued on the flange of the frame beam or column at each story. Strain gauge data were collected until the end of the test, in which the final loading PGA level was 1.6 g for the Kobe wave. The maximum tensile and compressive strains derived from the time-history strains, measured at the bottom, middle with an 800 mm height from the ground, and top with a 1600 mm height of the frame column, under the action of different seismic waves are shown in Figure 22. The strains in the column increased steadily with the increment of earthquake intensity levels. The envelope strain at the bottom of the frame column was the largest, followed by the middle, and the strain at the top was the smallest under the same PGA action, illustrating that the column base was the weakest compared to the components at the other locations of the column.

The maximum strains of the upper and lower flanges at the ends of the bottom and top beams, subjected to the different intensity levels, are shown in Figure 23. The discrepancy of strains between the bottom and top beams was not remarkable when PGA was less than 0.7 g. However, the maximum strain at the bottom beam was slightly higher than that at the top beam after PGA = 0.7 g, because the slopes of the strain curves at the bottom beam increased sharply. This reveals that the strength requirement of the bottom beam, in the practical design of the structure, should be given more attention compared to the top beam under the higher intensity level. In addition, the maximum strains at the upper and lower flanges of beams at each story were approximately equivalent and symmetrical, and this regularity of strain distribution was almost identical for the three seismic waves.

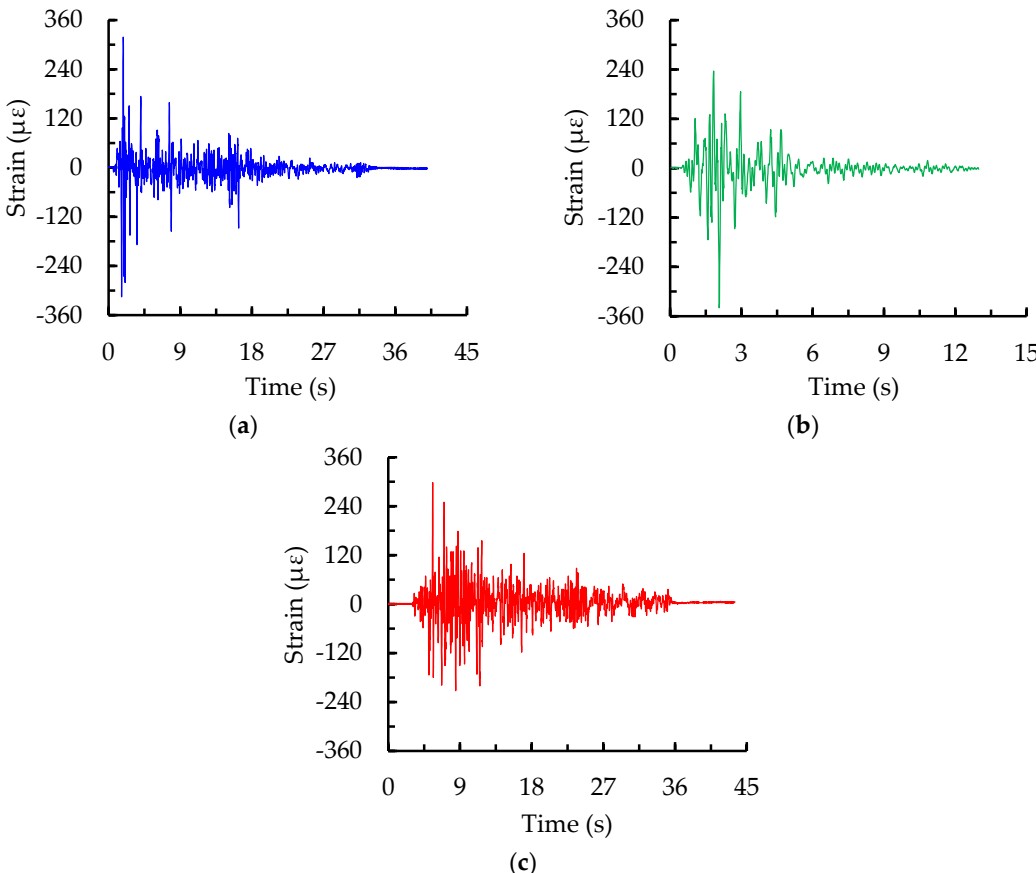

**Figure 21.** Time-history curves of strains at the bottom of the column under PGA of 1.0 g. (**a**) El Centro wave; (**b**) Kobe wave; (**c**) Taft wave.

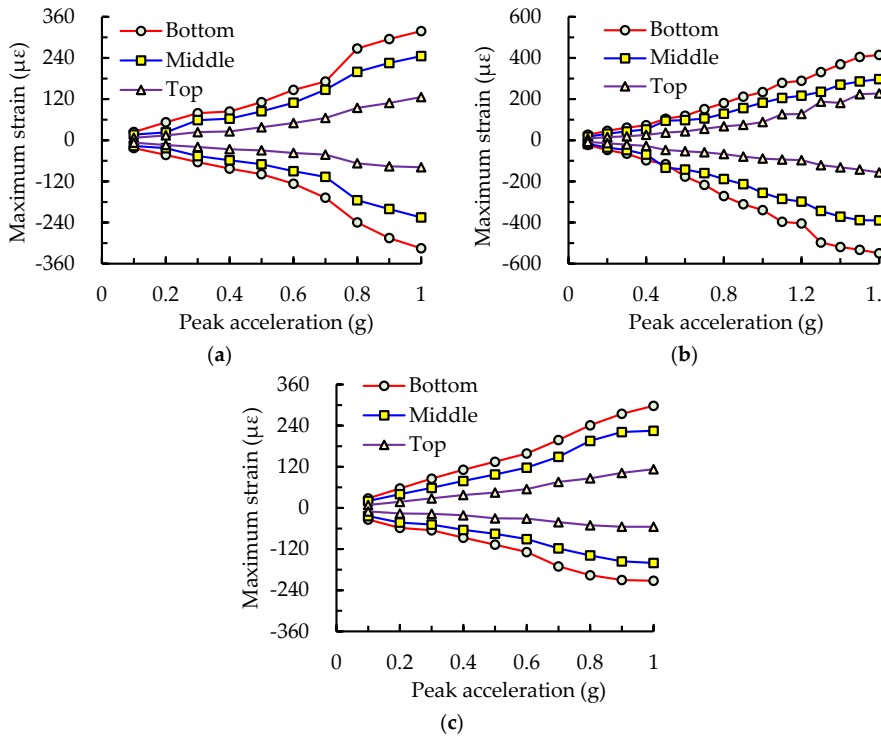

**Figure 22.** Maximum strains of the frame column subjected to the different intensity levels. (**a**) El Centro wave; (**b**) Kobe wave; (**c**) Taft wave.

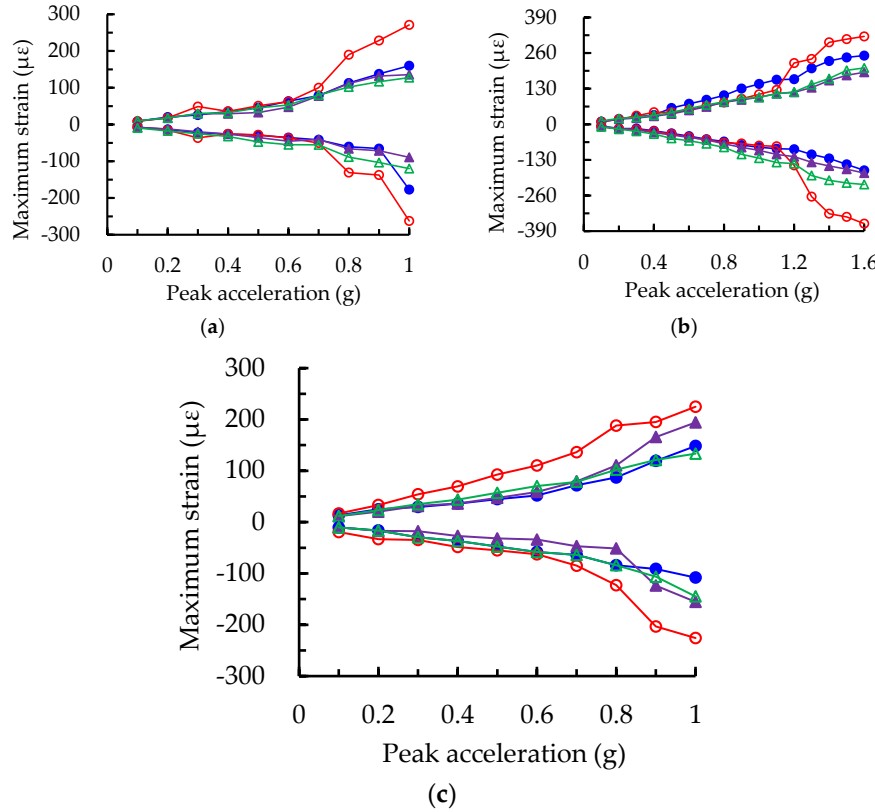

Legends: —●— Upper flange at bottom —○— Lower flange at bottom —▲— Upper flange at top —△— Lower flange at top

**Figure 23.** Maximum strains of the flange at the end of the frame beam subjected to the different intensity levels. (**a**) El Centro wave; (**b**) Kobe wave; (**c**) Taft wave.

It could be seen from Figures 22 and 23 that the maximum strains in the frame column and beam were, respectively, 550 and 363 με under the final loading PGA level of 1.6 g, approximately to rare earthquake intensity 9, which were far less than the measured yield strain of 1140 με obtained by the material test for the utilized Q235 steel. This shows that no yielding or failure occurs in the frame column and beam, and the main structural members continue to remain elastic. This phenomenon can be attributed to the fact that most of the hysteretic input energy is significantly dissipated by the LYP steel energy absorbers, thus reducing the force acting on the parent structure induced by the earthquake or wind loads.

### 3.6. Strain Responses of Energy Absorbers

Figure 24 presents the maximum strain responses of low-yield-point steel energy absorbers at the bottom floor under different loading intensity levels of three seismic waves. When the PGA of the earthquake record was less than 0.7 g, the line connecting the envelope strain points of energy absorbers at each level was approximately straight, and the strain discrepancies under the same intensity level for different seismic waves were quite small. However, the discrepancies became more significant after the increasing PGA was greater than 0.7 g, in agreement with the initially intense vibration of the overall specimen, accompanied by the visible displacement observed in the tests. The maximum tensile strain in the energy absorbers was 476 με, reaching and slightly exceeding its average yield strain of 470 με calculated from Table 5, when the PGA of the Kobe wave was 1.1 g. It reveals that the LYP plates gradually begin to dissipate the earthquake energy. With the further increase of loading intensity level, the envelope tensile strain of LYP energy absorbers became much larger and reached 718 με, far exceeding its yield strain, under PGA of 1.6 g nearly rare earthquake intensity 9 at the end of the test. A considerable plastic deformation was formed in the metallic energy absorbers, and its energy dissipation performance was gradually shown during the strong excitations.

It could be concluded, from the aforementioned analyses, that the frame columns or beams remained in the elastic stage with minor deformation, but energy absorbers significantly yielded and underwent permanent deformation. In other words, the structural failure of the steel frame retrofitted by energy absorbers primarily concentrated at the LYP plate, which utilized its superior ductility to form hysteretic bending plastic deformation, with an abundant amount of energy dissipation, so as to prevent the parent structure from damage under the severe earthquake.

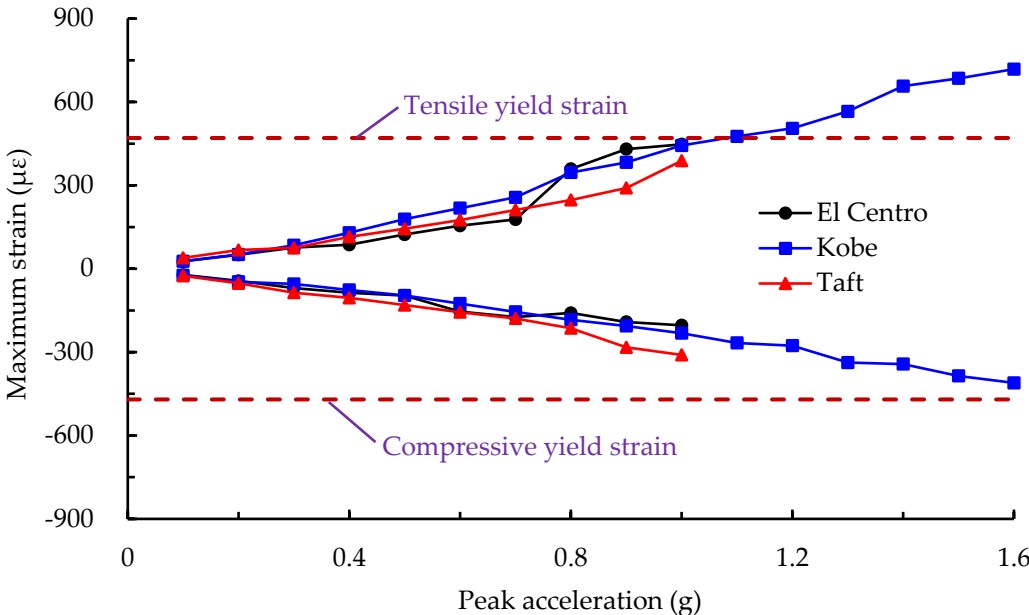

**Figure 24.** Maximum strains of LYP energy absorbers subjected to different intensity levels.

### 4. Conclusions

Several high-intensity earthquakes have occurred worldwide and caused enormous economic losses or deaths in recent years. With the aim of mitigating the detrimental effects induced by earthquake disasters, LYP steel, with excellent ductility capacity, can be potentially adopted as an effective energy dissipation device in structural systems such as high-rise buildings, long-span structures, and railway bridges. However, the inelastic dynamic responses, with repeated cyclic out-of-plane buckling and yielding of the LYP steel energy absorbers subjected to the earthquake action, is highly complex and difficult to predict accurately by any conventionally analytical or numerical method. The goal of this paper is to clearly demonstrate, by a rigorous experimental program, that the benefits of using energy absorbers, which behave just like a "fuse" in buildings, can significantly improve the seismic performance of structures. The insight into the dynamic behaviors of the test structure will enable a better guideline for seismic design under strong earthquakes.

In this study, a number of experiments were conducted to investigate the non-linear dynamic seismic responses of a quarter-scaled model of a steel frame retrofitted by LYP steel energy absorbers. The test specimens were subjected to a shaking table test with gradually increasing acceleration amplitudes, representing incremental earthquake intensities. The following main conclusions can be drawn from the experimental studies.

(1) Incorporation of the energy absorbers into high-rise buildings can significantly enhance the seismic performance of the structure, which provides a low-cost alternative solution for structural designers. The low-yield-point steel with superior hysteretic behavior should be primarily adopted as the material used in the energy absorbers, which enter the yield stage prior to the main members of the frame in order to provide energy dissipation when subjected to the earthquake action. Attention should be paid to the shape of the plate when designing the energy absorbers. The cross-sectional width of the LYP plate varies with the height, which is smallest at the middle and largest at both ends. The arc transition adopted at the cross-sectional change is an effective way to improve the local stress concentration.

(2) The measured initial fundamental natural frequency and initial damping ratio of the specimen were 15.68 Hz and 3.17% subjected to the first loading stage of 0.1 g, respectively, which remained essentially unchanged before PGA < 0.6 g. As the PGA increased from 0.7 g to 1.0 g, the natural frequency gradually decreased while the damping ratio slightly increased. These results indicated that structural stiffness degradation and slightly accumulated damage occurred. The results were in a good agreement with the experimental observations on the loosening of the high-strength bolts at column bases.

(3) The acceleration amplification factors $\beta$ under PGA = 1.0 g at the second floor were, respectively, 2.3, 2.42, and 2.63, subjected to the El Centro, Kobe, and Taft waves, while those at the first floor were 1.4, 1.32, and 1.78, respectively. The $\beta$ value increased with the height and exhibited a sharp increase before a certain input acceleration level, and then it gradually decreased with a further increasing acceleration.

(4) The maximum inter-story drift ratio of the test model was between 1/2310 and 1/1742 in the case of frequent earthquakes and was 1/233 in the case of rare earthquakes, which was significantly less than the allowable limit values given in China's seismic code. This clearly reveals that the structure possesses an extremely high safety reserve, to sufficiently satisfy the design requirements, and is remarkably suitable for applicability in high-intensity seismic fortification areas.

(5) The maximum strains in the frame column and beam under the final loading PGA level of 1.6 g were 550 and 363 $\mu\varepsilon$, respectively, which were less than the measured yield strain of 1140 $\mu\varepsilon$ from the material tests. On the other hand, the envelope tensile strain of the LYP energy absorbers reached 718 $\mu\varepsilon$, far exceeding its average yield strain of 470 $\mu\varepsilon$. This indicates that no yielding or failure occurs in the frame column and beam, and the seismic failure concentrates on the energy absorbers. It can be concluded that the failure mechanism of steel frame retrofitted by energy

absorbers is primarily located at the LYP plate. The retrofitted steel frame utilizes its ductility to develop hysteretic bending plastic deformation, unleashing an abundant amount of energy dissipation in order to prevent the parent structure from damage under a severe earthquake.

**Author Contributions:** Conceptualization, J.S., K.W. and S.K.; methodology, J.S. and W.C.; data curation, J.S., K.W. and Z.W.; formal analysis, J.S., K.W. and S.K.; investigation, J.S., K.W. and S.K.; resources, Z.W.; writing—original draft, J.S. and K.W.; writing—review and editing, S.K. and W.C.; visualization, J.S. and K.W.; supervision, S.K.; project administration, Z.W.; funding acquisition, J.S., S.K. and Z.W.

**Funding:** This work was financially supported by the National Natural Science Foundation of China (Grant No. 51308260), Jiangsu Overseas Visiting Scholar Program for University Prominent Young and Middle-Aged Teachers and Presidents (Grant No. 2018169), Science and Technology Program of Guizhou Province (Grant No. [2019]1288), and Innovation Group Major Research Project of Guizhou Education Department (Grant No. [2017]048).

**Acknowledgments:** The third author wishes to gratefully acknowledge the Japan Society for Promotion of Science (JSPS) for his JSPS Invitation Research Fellowship (Long-term), Grant No. L15701, at Track Dynamics Laboratory, Railway Technical Research Institute and at Concrete Laboratory, the University of Tokyo, Tokyo, Japan. The JSPS financially supports this work as part of the re-search project, entitled "Smart and reliable railway infrastructure". Special thanks to European Commission for H2020-MSCA-RISE Project No. 691135 "RISEN: Rail Infrastructure Systems Engineering Network" (www.risen2rail.eu) [43]. In addition, the collaboration and assistance from EU Cost Action TU1404 (Towards the next generation standards for service life of cement-based materials and structures) is highly appreciated.

**Conflicts of Interest:** The authors declare that there are no conflicts of interest regarding the publication of this paper.

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
