# Peer review of "Experimental Investigations into Earthquake Resistance of Steel Frame Retrofitted by Low-Yield-Point Steel Energy Absorbers"

_applsci, doi:10.3390/app9163299_

Round 1

Reviewer 1 Report

In the paper experimental shaking table tests on a two-story, one-span (1/4-scaled model) novel steel frame retrofitted by metallic bending energy absorbers subjected to 24 increasing levels of input earthquake records are presented.

The paper deals with an interesting topic in the field of Seismic Engineering which deserves to be deepened.  It is almost well written and presented, bust some improvements are required to be accepted for publication. The proposed suggestions are herein reported:

-          The use of structural fuses in the structural engineering field is a very consolidated technique. Low-yield devices for seismic applications were cited in a lot of European papers, i.e.

https://www.scopus.com/record/display.uri?origin=citedby&eid=2-s2.0-85061713122&ctoId=CTODS_1107503196&noHighlight=false&relpos=2

https://www.scopus.com/record/display.uri?eid=2-s2.0-80052889128&origin=reflist&recordRank=

https://www.scopus.com/inward/record.uri?eid=2-s2.0-84891900657&doi=10.1016%2fj.engstruct.2013.12.005&partnerID=40&md5=1e916b28b877c245dd20db4c8d6f562c

-          Design and use of passive control devices can be found in some papers, i.e.:         

https://www.scopus.com/record/display.uri?eid=2-s2.0-47549092965&origin=reflist&recordRank=

https://www.scopus.com/record/display.uri?eid=2-s2.0-67649227734&origin=reflist&recordRank=

https://www.scopus.com/record/display.uri?src=s&origin=cto&ctoId=CTODS_1107503196&stateKey=CTOF_1107503197&eid=2-s2.0-61749086414

https://www.scopus.com/record/display.uri?src=s&origin=cto&ctoId=CTODS_1107503196&stateKey=CTOF_1107503197&eid=2-s2.0-56249084950

https://www.scopus.com/record/display.uri?eid=2-s2.0-0035400914&origin=reflist&recordRank=

-          It should be specified the chemical content of used steels and also their provenance. Some comparison with the properties of usual LYP steels used in Japan and mentioned in some of the above papers could be done.

-          Some discussion on the difference of behaviour between scaled model and full-scale one should be provided.

-          Provisions on the possible use and design method of the examined device into high-rise buildings should be given.

-          English should be revised.

Reviewer 2 Report

The paper is concerned about the seismic performance of low yield steel retrofitted steel frames. The experimental setup is interesting; however it is quite limited in investigating the behavior of such frames. In the end, the authors struggle in drawing significant conclusions that can be generalized for the mentioned structures.

Significant improvement is required before the paper can be recommended for publication.

Some general comments:

The written language should be revised. Preferably language editing should be done by an external body.

The paper is too wordy compared to its content. Significant reduction in the text is suggested. The paper can be reduced to half its size without losing content.

The quality of all photographs should be increased.

The authors use too many abbreviations which reduces readability. Either these can be reduced or a nomenclature is needed.

Have you conducted any tests using the same frame without the energy absorbers? This would add significant value to the research in terms of comparison. If not, the authors are encouraged to compare their results with non-retrofitted frame tests from the literature.

Specific comments:

Line 37 – Refences are bad. Suggested to be removed.

Line 92 – Specify in the sentence that the code is Chinese code.

Figure 1 – The energy absorbers are suggested to be highlighted in the figure.

Figure 7 – missing

Line 206 – Specify what YHD & TST stands for. Please also elaborate on the types of displacement sensors, how they were installed and what they measure. Photos are encouraged.

Line 260 – What is the meaning of these frequencies? How are they obtained? Are they used in the rest of the paper? If not, why are they presented?

Table 7 – The change in damping ratio is not very high, especially if the uncertainty in damping estimation is considered. The yielded energy absorber should have increased damping significantly. How are the authors explain this?

Line 480 – The authors state that the beams are yielded. This shouldn’t happen under the presented strain levels. Also, this is contradictory to authors’ other comments (such as in Line 486). Please clarify this.

Line 487 – The stresses are already calculated using the elastic modulus, therefore under the assumption of linearity. It does not make sense to conclude elastic behavior using those stress values.

Figure 22 – The y-axis should be labeled “Maximum stress”

Figure 22 & 23 – What is the meaning of these stress values? Given there is no yielding (which is apparent from the strain measurements) and without comparisons with a regular unreinforced frame, I miss the point of these plots.

Line 508 – How was the maximum stress in the absorbers measured?

Line 533 – The word “excellent” is used extensively and carelessly throughout the paper.

Line 537 – Why is it hard to predict the response analytically? It is straightforward to model such a frame.  The authors are invited to explain this.

Line 533 – How about damping? The fact that damping is not changed significantly (as Table 7 suggests) implies that the overall energy dissipation capacity is almost unchanged.

Line 556 – Sharp decrease. Is this due to the yielding in the low-yield steel? Please explain.

Reviewer 3 Report

The paper is very interesting and well written. In the opinion of this reviewer it should be published with some minor suggestions.

In the introduction the authors state ”…metallic bending energy absorbers made of extremely 22 low-yield-point steel with the yield strength of only approximately 100 MPa”

But in fig. 8 the strain hardening of such material is greater than 100%, in fact the ultimate resistance is greater than 250 Mpa. Are the author sure that such high strain hardening has a positive effect on the seismic design and seismic behaviour?

It could be useful to add some considerations regarding this aspect. In fact in literature there are a lot of works dealing with dampers or special dissipative devices and all of them agree on the fact that it is better to have a low strain hardening material. In particular I suggest to add the following references:

Farzampour, A., Khatibinia, M., Mansouri, I.

Shape optimization of butterfly-shaped shear links using Grey Wolf algorithm (2019) Ingegneria Sismica, 36 (1), pp. 27-41.

Titirla, M., Katakalos, K., Zuccaro, G., Fabbroccino, F.

On the mechanical modeling of an innovative energy dissipation device (2017) Ingegneria Sismica, 34 (2), pp. 126-137.

As mentioned in the conclusion:

The goal of this paper is clearly to demonstrate that the benefits of using energy absorbers which work just like “fuse” in the building can significantly improve the seismic performance of the structure by the experimental method. The insight into the dynamic behaviours of the test structure will enable better guiding the seismic design under strong earthquakes.

This statement, from theoretical point of view, has been described in several papers, among which I suggest the following:

De Matteis, G., Brando, G., Caldoso, F., D'Agostino, F.

Seismic performance of dual steel frames with dissipative metal shear panels (2018) Ingegneria Sismica, 35 (2), pp. 124-141.

I suggest to add them in the references and to conclude that the present work could be seen as a prove of that theoretical previsions.

In addition, it could be interesting to clarify haw column and beams have been designed.

In fact. In the conclusions the authors state “no yielding or failure occurs in the frame column and beam, and the seismic failure concentrates on 569 the energy absorbers”

Did the author adopt a particular design criterion in order to assure that they remain in elastic range when energy absorbers dissipate seismic input energy?  At this aim I suggest to add the following papers in the references:

Dell'Aglio, G., Montuori, R., Nastri, E., Piluso, V.

A critical review of plastic design approaches for failure mode control of steel moment resisting frames (2017) Ingegneria Sismica, 34 (4), pp. 82-102.

The concept of using  “fuse” in the building because it can significantly improve the seismic performance of the structure has been adopted also in the following papers with reference to Reduced Beam Sections.

It could be interesting to recall them in the refernces:

Montuori, R., Sagarese, V.

The use of steel rbs to increase ductility of wooden beams (2018) Engineering Structures, 169, pp. 154-161.  

Montuori, R.

The influence of gravity loads on the seismic design of RBS connections (2014) Open Construction and Building Technology Journal, 8, pp. 248-261.

Some other useful references:

De Matteis, G., Brando, G., Caldoso, F., D'Agostino, F. Seismic
performance of dual steel frames with dissipative metal shear panels
(2018) Ingegneria Sismica, 35 (2), pp. 124-141.

Imanpour, A., Torabian, S., Mirghaderi, S.R. Seismic design of the
double-cell accordion-web reduced beam section connection (2019)
Engineering Structures, 191, pp. 23-38.

Mirzai, N.M., Attarnejad, R., Hu, J.W. Enhancing the seismic
performance of EBFs with vertical shear link using a new
self-centering damper (2018) Ingegneria Sismica, 35 (4), pp. 57-76.

Totter, E., Formisano, A., Crisafulli, F., Mazzolani, F. Seismic
upgrading of RC structures with only beam connected Steel Plate
Shear Walls(2018) Ingegneria Sismica, 35 (2), pp. 91-105.

Round 2

Reviewer 2 Report

The revised manuscript is very hard to read in this form. I kindly request the authors to submit a revised manuscript in its final form with only the additions and major changes are highlighted with color.

Round 3

Reviewer 2 Report

The paper is recommended for publication.

Author Response

Thank you very much for your acceptance. Revision for English improvement has been made.